# Chemical Mechanisms Underlying Sweetness Enhancement During Processing of Rehmanniae Radix: Carbohydrate Hydrolysis, Degradation of Bitter Compounds, and Interaction with Taste Receptors

**DOI:** 10.3390/foods14223932

**Published:** 2025-11-17

**Authors:** Wenting Zu, Jiasheng Wang, Jing Wang, Hongyue Wang, Liangliang Song, Yichen Li, Hongshuang Chi, Gaimei She, Hong Du

**Affiliations:** School of Chinese Materia Medica, Beijing University of Chinese Medicine, Beijing 102488, China; z18323118563@163.com (W.Z.); 15997531214@163.com (J.W.); wangjing000805@163.com (J.W.); bucm20200941402@163.com (H.W.); 20180222041@bucm.edu.cn (L.S.); asd2015392021@163.com (Y.L.); chihong1472580@163.com (H.C.)

**Keywords:** Rehmannia Radix, taste receptor interactions, carbohydrate hydrolysis, iridoid glycoside transformation, thermal processing, sweetness enhancement

## Abstract

Thermal processing is widely applied in food manufacturing to enhance flavor, but the mechanisms underlying taste transformation remain insufficiently understood. Rehmannia Radix, traditionally processed by steaming, undergoes a distinctive shift from bitterness to sweetness, serving as a representative model for flavor modulation during processing. In this study, sensory evaluation (n = 12), electronic tongue analysis, HPLC-based sugar and marker profiling across 17 batches, and molecular docking with representative human taste receptors were combined to investigate the mechanisms of taste transformation. The results showed that steaming markedly increased sweetness while reducing bitterness (*p* < 0.05). Chemical profiling revealed the hydrolysis of oligosaccharides into higher-sweetness monosaccharides (e.g., fructose (Fru) +15.99%, glucose (Glu) +8.90%) and substantial degradation of bitter iridoid glycosides (e.g., catalpol (Cat) −88%). In addition, the formation of 5-hydroxymethylfurfural (5-HMF) was identified as a processing marker. Molecular docking suggested that bitter glycosides in raw samples may interfere with sweet receptor activation and stimulate bitter receptors, whereas monosaccharide enrichment and Maillard products favored sweet receptor interactions, which may explain the observed sensory changes. Overall, these results clarify the chemical basis and receptor-level mechanisms of the bitterness-to-sweetness transition during steaming and identify markers useful for monitoring flavor changes in Rehmannia Radix.

## 1. Introduction

Rehmanniae Radix, derived from the fresh or dried tuberous roots of *Rehmannia glutinosa* Libosch (family Scrophulariaceae), is a well-known medicinal and edible plant that has demonstrated considerable potential in the development of modern functional foods. Owing to its unique sweet taste transformation during processing, it has been developed into a variety of health-oriented food products, including functional beverages such as “Dihuangbao”, kidney-tonifying Rehmannia-based congee, and “black gold” Rehmannia leaf tea formulated through specific techniques to eliminate bitterness. In addition, stachyose (Sta), a key oligosaccharide component in Rehmannia, is widely used as a high-quality prebiotic in yogurt and baked goods. Traditional products such as Rehmannia-based pickles and preserved fruits also highlight its value in regional specialty foods [1]. As a representative plant demonstrating the principle of “raw and processed medicinals have different action”, the sensory and functional characteristics of Rehmannia Radix are critically dependent on its processing techniques. The raw Rehmannia Radix (RRR) exhibits a characteristically bitter and astringent taste, limiting its food applications primarily to functional additives or nutraceutical supplements. Through traditional processing methods such as steaming and drying to produce Rehmannia Radix Praeparata (RRP), its organoleptic properties are significantly altered—the original bitterness is converted to a natural sweetness, enabling its widespread use as both a natural sweetener and multifunctional food ingredient [2].

Research indicates that humans have evolved a general sensory preference for sweetness while developing an aversion to bitterness. Most bitter compounds (e.g., quinine) trigger rejection responses by activating the T2R receptor family, whereas sweet compounds activate T1R2/T1R3 receptors, inducing pleasurable neural signals that significantly improve consumer acceptance [3,4,5]. The taste transformation of Rehmannia Radix through processing not only reduces sensory irritation when used as a food ingredient but also enhances patient compliance and product palatability through flavor optimization. Therefore, investigating the mechanisms of sweetness conversion during Rehmannia Radix processing can not only clarify the scientific basis of the traditional practice of “enhancing sweetness through processing” but also provide theoretical support for the standardization of modern processing techniques and the development of functional foods. To elucidate the chemical basis underlying this sweetness enhancement, it is necessary to summarize the major chemical constituents and their potential transformations during processing.

At present, a wide range of chemical constituents have been identified from Rehmanniae Radix, including iridoid glycosides, phenylethanoid glycosides, triterpenoids, flavonoids, amino acids, organic acids, and carbohydrates. Among them, the secondary metabolites, particularly iridoid glycosides such as catalpol (Cat), rehmannioside A (Reh A), rehmannioside D (Reh D), and ajugol (Aju), together with phenylethanoid glycosides such as echinacoside (Ech) and verbascoside (Ver), represent the major bioactive compounds responsible for its antioxidative, anti-inflammatory, and “heat-clearing” pharmacological activities. In addition, triterpenoids and flavonoids have also been detected as minor secondary metabolites exhibiting diverse biological activities [6,7,8]. Beyond these bioactive constituents, Rehmanniae Radix is also rich in carbohydrates, which are not only closely associated with its tonic effects but also play a key role in flavor transformation during processing. The carbohydrate fraction of Rehmanniae Radix mainly consists of soluble polysaccharides and oligosaccharides, such as Sta, raffinose (Raf), and manninotriose (Mnt), which are prone to hydrolytic cleavage under moist heat conditions, generating monosaccharides and disaccharides such as glucose (Glu) and fructose (Fru) [9,10], thereby contributing to increased sweetness. Moreover, the degradation of bitter iridoid glycosides also occurs during thermal processing [11]. These two reactions act synergistically to enhance the overall sweetness of Rehmanniae Radix, providing a plausible chemical basis for its characteristic bitterness-to-sweetness transformation. This unique coexistence of bitter iridoid glycosides and readily hydrolysable oligosaccharides distinguishes Rehmanniae Radix from most other traditional Chinese medicinal materials.

However, despite existing insights into the chemical transformations of Rehmanniae Radix, most current studies have primarily focused on the dynamic changes in pharmacologically active compounds and the optimization of processing techniques, whereas investigations addressing the mechanisms of sweetness enhancement from a flavor chemistry perspective remain limited. Quantitative analyses have shown that, during repeated steaming, the contents of iridoid glycosides such as Cat decrease markedly, while the levels of sweet sugars such as Fru increase significantly [12,13,14], indicating a close relationship between sweetness enhancement, the degradation of bitter compounds, and the accumulation of sugars. Although bitterness suppression is likely driven primarily by the reduction in bitter compounds, the concurrent increase in sugars such as Fru may further enhance overall sweetness through potential antagonistic interactions with bitter components. Additionally, Maillard reaction products, such as 5-hydroxymethylfurfural (5-HMF), have been shown to intensify sweetness perception through caramelization [15]. Nevertheless, most existing studies focus on the static quantification of individual components, without establishing a systematic correlation between dynamic chemical transformations and sensory evaluations. Moreover, the molecular mechanisms underlying the interactions between dynamically transformed compounds and taste receptors during processing remain insufficiently studied, leaving the synergistic chemical-sensory mechanism of sweetness enhancement yet to be fully elucidated.

Molecular docking has emerged as a critical tool for elucidating ligand–receptor interaction mechanisms, demonstrating substantial advantages in research on food flavor perception and the pharmacological properties of traditional Chinese medicine. By modeling the three-dimensional binding conformations between small molecules and taste receptor families, molecular docking can reveal key interaction features such as hydrogen bonding networks and spatial compatibility, thereby providing insights into the modulatory potential of compounds on taste signaling pathways. For instance, in studies exploring umami enhancement mechanisms in foods, molecular docking has shown that umami peptides interact stably with key residues of the umami receptor T1R1/T1R3 (Ser150, Ser256, Glu128) via hydrogen bonds, confirming that the structural flexibility and extended receptor-binding coverage of longer-chain peptides contribute to umami perception [16]. Similarly, molecular docking has been applied in optimizing traditional Chinese medicine processing methods, precisely identifying flavonoid binding sites and interaction modes with bitter taste receptors in the TAS2R family. In combination with physical barrier strategies and encapsulation effects, these findings provide molecular-level evidence to construct a systematic framework of “bitter compound–receptor interaction–pharmacological effects,” thereby guiding innovations in targeted bitterness reduction techniques [17,18]. To investigate the potential mechanisms by which key compounds formed during the processing of Rehmanniae Radix modulate taste perception, this study focused on representative human sweet and bitter taste receptors. The heterodimeric sweet receptor hT1R2/hT1R3 is widely recognized as the primary sensor for sugars and sweeteners, mediating sweetness perception through G protein-coupled signaling pathways. For bitterness perception, hT2R4 and hT2R14 (two members of the TAS2R family) have been extensively studied due to their broad ligand profiles and high sensitivity to bitter phytochemicals, particularly those derived from medicinal herbs. Therefore, hT1R2, hT1R3, hT2R4, and hT2R14 were selected as representative targets for molecular docking analysis in this study.

Therefore, the main purposes of this study are: (1) to systematically analyze the taste evolution characterized by sweetness enhancement and bitterness reduction during the processing of Rehmannia Radix through the integration of sensory evaluation and electronic tongue technology, thereby validating the traditional empirical principle of “sweetness enhancement through processing”; (2) to clarify the dynamic changes in oligosaccharide hydrolysis, monosaccharide enrichment, and bitter compound degradation using chemical analysis methods, thereby revealing the chemical mechanisms underlying the transition from bitterness to sweetness; (3) to elucidate the regulatory mechanisms of iridoid glycosides, phenylethanoid glycosides, and saccharides on sweet and bitter taste receptors through molecular docking simulations, thereby uncovering the molecular basis of taste modulation induced by processing.

## 2. Materials and Methods

### 2.1. Materials

#### 2.1.1. Samples

Seventeen batches of Rehmannia Radix samples were collected from Henan Province (Table 1), with assistance from Dong’e Ejiao Co., Ltd. (Liaocheng, Shandong, China). Professor Jingjuan Wang from Beijing University of Chinese Medicine identified them as the tuberous roots of *Rehmannia glutinosa* Libosch.

The raw Rehmanniae Radix (RRR) and Rehmannia Radix Praeparata (RRP) slices were prepared in our laboratory in accordance with the processing methods specified in the Chinese Pharmacopoeia (2020 edition), and key processing parameters were recorded to ensure reproducibility. The Rehmannia material used in this study was obtained as partially processed medicinal roots that had already undergone the initial post-harvest handling at the production site, including removal of impurities and controlled drying to about 80% dry mass, as required by the Pharmacopoeia. For preparation of RRR, this semi-dried material was weighed and rehydrated by adding approximately 20% (*w*/*w*) water to achieve uniform softening and facilitate slicing. The rehydrated roots were then cut into thick slices (approximately 2–4 mm, consistent with the general slicing specification in the Pharmacopoeia) and dried in a hot-air oven at 60 °C for approximately 4 h, until they reached a pliable, slightly sticky state with a dark brown to brownish-black surface. The resulting slices were defined as RRR.

The RRP slices were prepared from the same source material. For preparation of RRP, the material was thoroughly mixed with yellow rice wine at approximately 40% (*w*/*w*) relative to the root mass, then steamed in a closed vessel for about 24 h until the roots became uniformly soft, blackened, and mucilaginous throughout. The processed roots were removed and allowed to rest in air until the surface mucilage became tacky rather than wet, then cut into thick slices and further dried in a hot-air oven at 60 °C for approximately 3 h to obtain the final processed slices, which exhibited a glossy black surface and a pliable, slightly adhesive texture characteristic of RRP.

The above processing procedures were carried out in strict accordance with the requirements of the Chinese Pharmacopoeia. All batches of RRR and RRP were then prepared under the same controlled processing conditions to ensure batch-to-batch consistency and experimental reproducibility.

Both RRR and RRP samples comprised 17 individual specimens, labeled as R1–R17 and P1–P17, respectively. All samples were stored at 4 °C before further analysis.

#### 2.1.2. Reagents

Standard reference compounds, including Cat, Reh D, Reh A, Aju, Ech, cistanoside A (Cis A), Ver, 5-HMF, Fru, Glu, sucrose (Suc), melibiose (Mel), Raf, Mnt, and Sta, were purchased from Beijing Better & Renomed Biotechnology Co., Ltd. (Beijing, China), all with a purity of ≥98%. HPLC-grade acetonitrile was obtained from Thermo Fisher Scientific (Shanghai, China), and HPLC-grade phosphoric acid was acquired from Macklin Biochemical Co., Ltd. (Shanghai, China). Analytical-grade methanol was supplied by Beijing Chemical Works Co., Ltd. (Beijing, China), and Wahaha purified water was purchased from Hangzhou Wahaha Group Co., Ltd. (Hangzhou, China). Yellow rice wine (GB/T 13662 [19], Dongfeng Huadiao Aged Shaoxing Wine, Shaoxing Kuaijishan Shaoxing Wine Co., Ltd., Shaoxing, Zhejiang, China; batch no. 20230915) was used as the processing medium during the preparation of RRP.

### 2.2. Sensory Analysis

According to the Chinese Standard GB/T 16291.1-2012 [20] and with reference to supplementary sensory evaluation protocols, including GB/T 10220-2012 [21], GH/T 1408-2022 [22], DB42/T 2253-2024 [23], and T/HBLS 0015-2023 [24], 12 sensory panel members (seven males and five females, aged 20–30 years) were recruited from the School of Chinese Materia Medica, Beijing University of Chinese Medicine. The sample size was determined according to GB/T 39625-2020 [25], which recommends 12–15 panel members for single-sample evaluations using direct scoring methods. Before the experiment, all participants underwent systematic training and qualification assessments covering six basic taste modalities: sweet, sour, bitter, salty, umami, and astringent.

During training, standard solutions were employed as taste reference standards for each of the six basic taste modalities, including 1.6% Suc (sweet), 0.1% citric acid (sour), 0.01% berberine hydrochloride (bitter), 0.5% sodium chloride (salty), 0.1% monosodium glutamate (umami), and 0.05% tannic acid (astringent). In the taste-matching test, each panelist was required to correctly identify the taste type of coded standard solutions and match them to corresponding reference samples. An accuracy rate of at least 80% was required to qualify for participation. In addition, all panelists completed a taste intensity discrimination test using citric acid solutions at four concentrations (0.10, 0.15, 0.22, and 0.34 g/L), which were arranged in ascending order of perceived sourness to verify panelists’ sensitivity and discrimination ability. All panelists passed the evaluations and proceeded to participate in the formal sensory assessment. Ethical approval for the study was granted by the Ethics Committee of Beijing University of Chinese Medicine (Approval No. 2025BZYLL0306), and all participants provided informed consent.

In the formal sensory evaluation, five basic taste descriptors (sweet, sour, bitter, salty, umami) were used to quantify the taste attributes of each sample. Astringency was recorded qualitatively as part of the aftertaste description and was not included in the quantitative scoring. The RRR slices from batch R2 were used as the reference sample, and a 7-point intensity scale was employed to assess the intensity of each taste attribute (Table 2), wherein the reference sample was assigned a baseline score of 0 for all attributes. Astringency was not scored but was described qualitatively during aftertaste evaluation. The other samples were scored on a relative scale ranging from −3 (much weaker) to +3 (much stronger), depending on the perceived intensity compared to the reference [26]. Panelists independently evaluated each taste attribute, and the scores were recorded in the sensory evaluation form.

For each test, approximately 1.0 g of each RRR or RRP sample was placed in a tasting cup labeled with a three-digit random code and presented in randomized order. Sensory blinding was rigorously maintained throughout all testing sessions: Samples from each batch were labeled with randomized three-digit codes, ensuring both panelists and session coordinators remained blinded to batch identities. The coding system was maintained consistently across all batches to prevent any bias related to batch identification. Taste evaluation was performed through direct chewing: each panelist chewed the sample in the mouth for 5–7 s, using oral sensory perception to assess the intensity of each taste attribute. Scores were recorded immediately after chewing. Each sample was evaluated three times, and the average score was used as the final result. To prevent residual flavors from affecting subsequent evaluations, panelists rinsed their mouths with purified water at least five times after each sample and rested for 15 min before proceeding to the next group. These measures ensured the accuracy and consistency of sensory assessments.

### 2.3. Electronic Tongue Analysis

The electronic tongue ASTREE II (Alpha MOS Company, Toulouse, France) was employed in this research to characterize the taste of RRR and RRP samples, with slight modifications based on a previously reported method [27]. The instrument was equipped with the sixth-generation sensor system, which included seven sensors: AHS, ANS, SCS, CTS, NMS, PKS, and CPS. These sensors were capable of detecting five basic taste attributes: sourness, sweetness, bitterness, saltiness, and umami, as well as two composite taste parameters. Meanwhile, an Ag/AgCl electrode was used as the reference electrode.

The sample preparation for electronic tongue analysis was carried out using an ultrasonic-assisted water extraction method to ensure efficient and reproducible extraction of sensory-active compounds. Specifically, 1.5 g of RRR or RRP powder was accurately weighed and mixed with 100 mL of drinking water. The mixture was soaked for 30 min to facilitate hydration, followed by ultrasonic extraction under controlled conditions (600 W, 40 kHz) for 30 min. After extraction, the sample was centrifuged at 6000 rpm for 10 min. The resulting supernatant was filtered through qualitative filter paper to obtain the final aqueous extract for analysis. Subsequently, 25 mL of the extract was transferred into a 50 mL beaker specifically designed for the electronic tongue system. The detection time was set to 120 s, and the analysis time was set to 180 s. After each measurement, the system was rinsed with distilled water for 2 min to remove interference. Each sample was analyzed nine times. To ensure sensor stability and minimize the influence of initial fluctuations or signal drift, the first five measurements were discarded, and the average of the final four readings was used as the representative result. All measurements were performed at a controlled temperature of 25 ± 1 °C.

### 2.4. Chemical Quantification Methods

#### 2.4.1. Carbohydrate Analysis

The sample preparation was carried out using an ultrasonic extraction method to achieve efficient and reproducible extraction of target compounds. Prior to extraction, slices of RRR and RRP were ground into coarse powder using an LG-01 high-speed Chinese herbal grinder (Ruian Baixin Pharmaceutical Machinery Co., Ltd., Rui’an, Zhejiang, China). The particle size of the coarse powder was controlled according to the definition in the Chinese Pharmacopoeia (2020 edition), namely that all powder passes through a No. 2 sieve (aperture 850 ± 29 μm, 24 mesh) while not more than 40% passes through a No. 4 sieve (aperture 250 ± 9.9 μm, 65 mesh). Subsequently,0.1 g of RRR or RRP coarse powder was accurately weighed and mixed with 25 mL of 70% methanol solution. The mixture was then subjected to ultrasonic extraction under controlled conditions (600 W, 40 kHz) for a duration of 30 min to maximize the dissolution of analytes. Following extraction, the sample was centrifuged at 9000 rpm for 10 min. Then, the supernatant was collected and filtered through a 0.22 μm membrane, yielding the final sample solution for further analysis.

The analytical method was optimized based on a previous study to improve detection accuracy and reliability [28]. An Agilent 1260 high-performance liquid chromatography (HPLC) system, coupled with an Allchrom 6100 evaporative light-scattering detector (ELSD), was utilized for the qualitative and quantitative analysis of the samples. Chromatographic separation was performed using an ItolSep ES Carbohydrate analytical column (250 mm × 4.6 mm, 5 μm; MORHCHEM Technologies Inc., Ontario, CA, USA). The mobile phase consisted of acetonitrile (A) and water (B), with a flow rate of 1.0 mL/min. The column temperature was maintained at 30 °C, and the injection volume was set at 10 μL. In the ELSD detection process, the drift tube temperature was set at 100 °C, and the carrier gas (high-purity nitrogen) was maintained at a constant flow rate of 2.0 L/min to optimize signal sensitivity. The gradient elution program was as follows: 0–10 min, 75% A; 10–15 min, 75% to 65% A; 15–25 min, 65% A (isocratic); and 25–27 min, 65% to 75% A.

#### 2.4.2. Characteristic Spectrum and Marker Compound Analysis

This section aimed to obtain the characteristic chromatographic fingerprint of RRR and RRP at the level of soluble low-molecular-weight constituents, to screen chromatographic peaks that underwent pronounced changes during processing, and to use these peaks as the initial pool of potential marker compounds for subsequent qualitative identification and quantitative analysis. This procedure was intended to provide a chemical basis for interpreting the enhanced sweetness and reduced bitterness observed in RRP relative to RRR. To this end, ultrasound-assisted extraction was used to enrich soluble low-molecular-weight constituents, and the resulting extracts were subsequently subjected to high-performance liquid chromatography (HPLC) for separation and detection, as described below. Specifically, 1.0 g of accurately weighed RRR or RRP coarse powder was mixed with 25 mL of 25% methanol solution. The mixture was then subjected to ultrasonic-assisted extraction under precisely controlled conditions, with a power output of 600 W and a frequency of 40 kHz, for 30 min. After extraction, the sample was centrifuged at 9000 rpm for 10 min. Then, the resulting supernatant was collected and filtered through a 0.22 μm membrane, yielding the sample solution for chromatographic analysis.

The analytical method was adapted with minor modifications based on a previous study to improve separation efficiency and peak resolution of key marker compounds under our laboratory conditions [29]. A Waters 2996-PDA high-performance liquid chromatography (HPLC) system (Waters Corporation, Milford, MA, USA) was employed for analysis. The chromatographic separation was carried out using an Agilent 5 HC-C18 analytical column (250 mm × 4.6 mm, 5 μm) to ensure efficient separation of target analytes. The mobile phase consisted of acetonitrile (A) and 0.01% phosphoric acid aqueous solution (B), with a flow rate of 1.0 mL/min. The column temperature was controlled at 30 °C, and the injection volume was set to 10 μL. To achieve effective separation, the gradient elution program was optimized and implemented as follows: 0–15 min, 2% to 5% A; 15–20 min, 5% A (isocratic); 20–30 min, 5% to 12% A; 30–34 min, 12% to 18% A; 34–52 min, 18% to 35% A. The detection wavelength was set at 203 nm to ensure optimal sensitivity for the target compounds.

### 2.5. Molecular Docking Methodology

A comprehensive search of the Protein Data Bank (PDB) was conducted to identify experimentally determined structures for hT1R2, hT1R3, hT2R4, and hT2R14. No high-resolution crystal structures were available for these specific human taste receptors at the time of analysis, necessitating the use of homology modeling approaches. The amino acid sequences of the sweet taste receptors (hT1R2, hT1R3) and bitter taste receptors (hT2R4, hT2R14) were downloaded from the Protein Database (NCBI, http://www.ncbi.nlm.nih.gov/protein/, accessed on 10 March 2025) with their respective IDs being NP_689418.2, NP_689414.2, NP_058640.1, and NP_076411.1. The protein structures were then modeled online using the I-TASSER server for protein structure and function prediction. Subsequently, the obtained models were evaluated using the SAVESv6.1 server, and the best model for each receptor was selected. Finally, the DoGSiteScorer tool (https://protein.plus, accessed on 15 March 2025) was employed to predict binding pockets in the protein structures online.

The molecular structures of a series of small-molecule compounds (Figure 1) and their 3D structure files in .sdf format were obtained from the PubChem compound database (https://pubchem.ncbi.nlm.nih.gov/, accessed on 21 March 2025). For small molecules that were only available in 2D structure format, the Avogadro (1.2.0) software was used to convert them into 3D structures, followed by energy minimization using the MM2 force field, and the optimized structures were then exported in .sdf format. The Open Babel (3.1.0) software was employed to convert the downloaded .sdf files into .pdb format. Subsequently, AutoDock Tools 1.5.7 was used for ligand preprocessing, which included the addition of all hydrogen atoms, defining the molecules as ligands, detecting torsional bonds, and finally exporting them as .pdbqt files. The best receptor models obtained through homology modeling were imported into AutoDock Tools 1.5.7 for protein preprocessing, where all water molecules were removed, hydrogen atoms were added, and the proteins were set as receptors before being exported in .pdbqt format.

Molecular docking was performed using AutoDock Vina 1.1.2, and all visualization tasks were carried out using PyMOL 3.1.0.

### 2.6. Data Analysis

Statistical analysis was performed using SPSS Statistics 20 software (IBM, New York, NY, USA), and data visualization was conducted using Origin 2024 (OriginLab, Northampton, MA, USA). All experiments were conducted in triplicate. Data normality was assessed using the Shapiro–Wilk test. For data with normal distribution, independent-sample t-tests were used for two-group comparisons, and one-way analysis of variance (ANOVA) followed by Tukey’s post hoc test was applied for comparisons among more than two groups. For data not normally distributed, Kruskal–Wallis tests were performed, and Mann–Whitney U tests were used for pairwise comparisons. A *p*-value of less than 0.05 was considered statistically significant.

## 3. Results and Discussion

### 3.1. Sensory Taste Differences Between RRR and RRP

Sensory evaluation (Figure 2A) revealed significant taste differences between RRR and RRP. The sweetness intensity of RRP was significantly higher than that of RRR (*p* < 0.05), while its bitterness intensity was significantly lower (*p* < 0.05). This finding aligns with traditional processing theory, which characterizes RRP as having a pronounced sweetness comparable to sugar. In addition, RRP exhibited a slightly higher level of sourness compared to RRR, which may be attributed to the generation of acidic compounds during thermal degradation of sugars and early-stage Maillard reactions. Studies have shown that organic acids can be formed during the intermediate stages of the Maillard reaction, which may contribute to sour taste perception [30]. During the sensory evaluation, the participants did not perceive saltiness or umami, which may result from the fact that the concentrations of relevant components, such as glutamic acid and sodium ions, in Rehmanniae Radix were below the human taste threshold [31].

Regarding aftertaste descriptions, RRR was primarily associated with “bitter, astringent, and slightly sweet”, whereas RRP was described as “sweet, slightly sour, and slightly bitter”. This taste profile may be related to concentration-dependent taste interactions, which is consistent with studies suggesting that sweetness, sourness, and bitterness often exhibit mutual suppression (Figure 2B) [32,33]. The higher bitterness in RRR can be attributed to its abundant bitter components, such as iridoid glycosides and phenolic compounds, which markedly suppress sweetness perception through both peripheral receptor antagonism and central nervous system inhibition. However, the processing of RRP generates a large amount of sweet compounds, thereby suppressing the expression of both sourness and bitterness. As sweet compounds gradually dissipate in the oral cavity due to salivary interaction and enzymatic hydrolysis, trace amounts of sour and bitter components may re-emerge due to the release from sweetness suppression [34,35]. These findings show a clear sensory transformation from RRR to RRP and point to the need for further analyses to explain the mechanisms behind this change.

### 3.2. Electronic Tongue Taste Differences Between RRR and RRP

The electronic tongue analysis revealed significant taste differences between RRR and RRP. Figure 3A,B show radar plots of sensor responses for all batches of RRR (R1–R17) and RRP (P1–P17), respectively (Figure 3A,B). Compared to RRR, RRP exhibited a marked enhancement in sweetness-related sensor signals (CTS) and a notable reduction in bitterness-related sensor signals (PKS) (Figure 3A,B). This result was consistent with sensory evaluation findings, confirming that the processing technique enhances the sweet taste characteristics of Rehmannia Radix.

Furthermore, OPLS-DA analysis further supported the distinction between the taste profiles of RRR and RRP (Figure 3C). The two sample groups formed two clearly separated clusters in the score plot, suggesting that processing induced a systematic shift in the overall taste profile. To ensure the robustness of the model, a permutation test (n = 200) was conducted. The results showed that all permuted Q^2^ values were lower than the original Q^2^ value. The R^2^ and Q^2^ intercepts were below 0.3 and 0, respectively, indicating that the OPLS-DA model was reliable and not overfitted (Figure 4A).

Additionally, the electronic tongue detected a decrease in umami and saltiness in RRP compared to RRR. However, in the sensory evaluation, participants did not clearly perceive these two taste attributes. This discrepancy may be attributed to the high sensitivity of the electronic tongue sensors (with detection limits reaching 10^−6^ mol/L) and the threshold limitations of human taste perception [36,37].

The reduction in umami and saltiness may result from the loss of water-soluble compounds during the processing of Rehmannia Radix [38].

### 3.3. Chemical Analysis

#### 3.3.1. Carbohydrate Compounds

Based on HPLC-ELSD chromatographic analysis (Figure 5A–C), significant differences in seven carbohydrate compounds were detected during the transformation of RRR into RRP (Figure 5D). Specifically, the contents of Fru, Glu, Mel, and Mnt increased by 15.99%, 8.90%, 4.58%, and 30.49%, respectively, while Suc, Raf, and Sta decreased by 13.23%, 9.42%, and 43.83%, respectively (Table 3). The absolute quantitative values of these sugars are provided in the Appendix A). This phenomenon could be attributed to the hydrolysis of oligosaccharides at high temperatures. Sta can be hydrolyzed into Mnt and Fru, while Raf can degrade into Mel, Fru, and Suc, and Suc can be broken down into Fru and Glu. Notably, Mnt and Mel do not undergo hydrolysis under high-temperature conditions. Consequently, the contents of Suc, Raf, and Sta significantly decreased, whereas Mnt, Mel, and monosaccharides increased in RRP (Figure 6) [39].

The sweetness value (I) is an important indicator for measuring the sweetness intensity of carbohydrates and is conventionally defined relative to Suc (I = 1). Due to differences in their chemical structures, various carbohydrates exhibit significantly different sweetness values. The transformation of carbohydrate components provides a scientific basis for the enhanced sweetness of RRP.

Fru (I = 1.5) possesses high sweetness and can quickly elicit sweet taste perception, significantly enhancing sweetness intensity [40,41]. The hydrolysis of Suc into equimolar amounts of Fru and Glu (I = 0.7) results in a net increase in sweetness [42]. Furthermore, Fru and Glu can participate in the Maillard reaction during processing, leading to the formation of aromatic compounds with roasted notes such as 5-HMF, which indirectly enhance caramel-like flavors and prolong sweetness perception [43,44]. Additionally, the reduction in Sta (I = 0.22) and Raf (I = 0.22–0.3) may decrease the “dilution effect” caused by low-sweetness oligosaccharides, thereby preventing them from competitively occupying sweet taste receptor binding sites. This dynamic transformation of carbohydrates scientifically explains the intrinsic chemical logic behind the traditional concept of “processing enhances sweetness”. The OPLS-DA model (Figure 5E) further confirmed the distinction between RRR and RRP, with good model fit and predictive performance supporting the reliability of the result (Figure 4B). In addition, VIP analysis (Figure 5F) identified Glu, Suc, Raf, Mnt, and Sta (VIP > 1) as the key differential carbohydrates distinguishing RRR from RRP.

#### 3.3.2. Characteristic Spectrum and Quantification of Marker Compounds

The characteristic spectrum constructed using median vector multipoint correction revealed 12 characteristic peaks in RRR and 13 in RRP (Figure 7). Through comparison with reference standards (Figure 8A), Peak 5 (Cat), Peak 7 (Reh D), Peak 8 (Reh A), Peak 9 (Aju), Peak 10 (Ech), Peak 11 (Cis A), Peak 12 (Ver), and Peak 13 (5-HMF) were successfully identified. Among them, 5-HMF (Peak 13) was detected only in RRP, serving as a signature product of the Maillard reaction during processing. Its formation results from the condensation reaction between monosaccharides (e.g., Glu) derived from polysaccharide hydrolysis and amino acids [45]. The Maillard reaction is known to generate a complex array of compounds with diverse taste characteristics, including sweet, bitter, umami, and roasted flavors. However, due to the high reactivity and structural variability of many intermediates, comprehensive profiling is analytically challenging. In this study, 5-HMF was selected as a representative marker because of its high yield, chemical stability, and reported contribution to caramel-like and sweet-associated flavors.

Quantitative analysis showed that after processing, the contents of seven iridoid glycosides and phenylethanoid glycosides (Cat, Reh D, Reh A, Aju, Ech, Cis A, and Ver) significantly decreased (*p* < 0.05) (Figure 8B), with an average reduction ranging from 41.77% to 88.15% (Table 4). This phenomenon is closely related to the hydrolysis mechanism of iridoid glycosides under moist-heat conditions. The high-temperature and high-humidity environment promotes glycosidic bond cleavage, leading to the formation of aglycones or secondary metabolites [46,47]. Additionally, ethanol in rice wine enhances plant cell membrane permeability, facilitating the dissolution of glycosides from cells. Furthermore, ethanol molecules form hydrogen bonds with aglycones, accelerating the hydrolysis of glycosides into secondary metabolites [48]. Notably, the difference in the degradation rates of Reh D (61.96%) and Reh A (52.17%) may be attributed to their glycosyl substitution sites. The glucosyl group at C-7 of Reh D is more sensitive to moist-heat conditions, whereas the acetyl group at C-6 of Reh A may slow down hydrolysis [49,50]. The OPLS-DA model further supported the significant chemical differences between RRR and RRP (Figure 8C), consistent with the observed glycoside degradation. The model showed reliable predictive ability in distinguishing the two groups (Figure 4C). In addition, VIP value analysis indicated that Reh A, Reh D, Aju, and Ech (VIP ≥ 1) are key marker compounds distinguishing the two (Figure 8D).

Studies have shown that most iridoid glycosides exhibit bitterness [51,52]. Accordingly, their significant reduction during processing not only diminishes the inherent bitterness and astringency of RRR but also increases the perceptibility of sweetness in RRP, consistent with the sensory and electronic tongue results. Thus, the sweetness enhancement observed after processing is not attributable to sugar changes alone; the reduction in bitter constituents is also an important contributor.

### 3.4. Molecular Docking

The receptor models obtained by homology modeling were assessed using ERRAT scores, C-score, and Ramachandran plots to determine their structural reliability. The results demonstrated that the ERRAT scores of all selected receptor models were greater than 80, indicating a high degree of atomic spatial reliability (Table 5). For each receptor, the best model (Model I) exhibited a C-score ranging from −0.63 to 0.40, and Model I had a TM-score greater than 0.6 (Table 6). Additionally, Ramachandran plots revealed that more than 97% of amino acid residues were located in favored regions (Figure 9). Overall, these results confirmed that the selected receptor models (Figure 10) were structurally reliable and suitable for subsequent molecular docking analysis.

The docking results (Figure 11) suggested that the potential binding sites of small molecules with the sweet taste receptors (hT1R2 and hT1R3) were located in the VFTM domain of the receptor, specifically in the intermediate cavity formed by the two lobes of the extracellular structure. This observation is consistent with previous studies reporting the binding sites of different ligands to sweet taste receptors [53]. Since the sweet taste receptor subtypes hT1R2 and hT1R3 can independently transmit sweet taste signals and produce an additive effect, this study employed the sum of the lowest binding energies (As) of the two subtypes to quantify the overall interaction between ligands and sweet taste receptors [54]. Similarly, the sum of the lowest binding energies (Ab) of the bitter taste receptor subtypes hT2R4 and hT2R14 was used to assess the comprehensive interaction between ligands and bitter taste receptors [55].

The binding energy analysis (Table 7) indicates that iridoid glycosides and phenylethanoid glycosides have relatively strong potential binding affinities with sweet taste receptors (hT1R2/hT1R3), suggesting a possibility of competitive interaction with sweet compounds that might inhibit sweet receptor activation. These compounds also show strong binding with bitter taste receptors (hT2R4/hT2R14), likely contributing to bitterness perception. However, it should be emphasized that molecular docking provides a theoretical prediction of binding potential rather than direct evidence of physiological receptor activation or inhibition. The observed correlation between reduced iridoid glycoside content and enhanced sweetness suggests a potential causal relationship, whereby decreased levels of these bitter compounds may reduce competitive inhibition on sweet receptors and diminish bitter receptor activation. Meanwhile, the increased levels of sweet compounds enhance the activation of sweet receptors. Collectively, these biochemical changes contribute to explaining the decrease in bitterness and the increase in sweetness observed in processed Rehmanniae Radix.

The hydrogen bonding pattern further supports the above hypothesis (Table 8). Hydrogen bond analysis revealed that iridoid glycosides and phenylethanoid glycosides exhibited significant overlap with monosaccharides and oligosaccharides at the binding sites of sweet taste receptors. For example, Aju and Cat competed with Suc, Mel, Raf, and Sta for binding to ILE-306 of hT1R2 and THR-305 and HIS-388 of hT1R3. This predicted competitive binding at overlapping receptor sites could reduce sweet receptor activation by sugar molecules through direct competition for the same binding domains. Chemical analysis demonstrated that after RRR was processed into RRP, the contents of iridoid glycosides and phenylethanoid glycosides were significantly reduced. This reduction would be expected to result in both decreased competitive binding ability and bitter taste activation, further contributing to the enhanced sweetness of RRP.

Additionally, monosaccharides such as Glu and Fru formed multiple stable hydrogen bonds with sweet taste receptors (e.g., ASN-460 and HIS-311 of hT1R2), demonstrating strong binding capability, which is consistent with the specific recognition mechanism of sweet taste receptors for monosaccharides reported in previous studies [56,57]. Oligosaccharides, such as Mel and Mnt, exhibited a greater number of hydrogen bonds with sweet taste receptors, suggesting that oligosaccharide molecules may contribute to a more stable binding pattern at receptor sites. Therefore, the increased content of oligosaccharides may also contribute to sweetness enhancement [58].

As a newly generated compound in processed RRP, 5-HMF exhibited a binding energy of −10 kcal/mol for sweet taste receptors, which was slightly weaker than that of other non-sugar compounds. The computational modeling suggests potential bitter taste antagonistic effects through predicted interactions with bitter taste receptors. However, the physiological significance of these predicted interactions requires further investigation, as the actual concentrations of 5-HMF in oral cavity during consumption and its bioavailability for taste receptor interaction remain to be determined in future studies.

It should be noted that this study has some limitations. First, molecular docking provides theoretical predictions of compound-receptor interactions and does not directly demonstrate receptor activation, which requires experimental validation. Second, the study relies on in vitro chemical analysis and sensory evaluation without in vivo confirmation, and the translation to actual taste perception involves complex physiological processes. Despite these limitations, this work integrates sensory evaluation, chemical profiling, and computational modeling to provide valuable insights into the chemical and receptor-level mechanisms underlying sweetness enhancement in Rehmannia Radix.

## 4. Conclusions

This study advances a mechanistic explanation for the sweetness shift in Rehmannia Radix Praeparata (RRP). Before processing, raw Rehmannia Radix (RRR) contains abundant carbohydrates and small-molecule constituents, including iridoid and phenylethanoid glycosides. During moist-heat processing, soluble polysaccharides and oligosaccharides hydrolyze to monosaccharides, while Maillard chemistry increases 5-HMF and bitter-relevant glycosides undergo thermal degradation. Collectively, these changes reduce the bitterness of RRP and increase its perceived sweetness, in agreement with the sensory and electronic-tongue results. Molecular docking provides predictive insights, indicating that sweet- and bitter-related ligands may competitively occupy taste-receptor binding sites; processing-induced compositional changes may enhance activation of sweet-taste receptors while attenuating activation of bitter-taste receptors. By integrating sensory evaluation, chemical analysis, and computational modeling, this work elucidates the mechanistic basis of taste transformation during thermal processing.

These findings not only advance the fundamental understanding of how traditional processing alters taste profiles in medicinal plants but also propose a multidimensional mechanistic framework linking carbohydrate hydrolysis, bitter-glycoside degradation, and receptor-level interactions. This integrative perspective provides a testable hypothesis to interpret the bitterness-to-sweetness transformation during processing. Moreover, the results offer mechanistic insight that may inform future research aimed at enhancing the sensory quality of RRP and guide the development of plant-based ingredients with enhanced sweetness and reduced bitterness, supporting the formulation of naturally sweetened functional foods.

## Figures and Tables

**Figure 1 foods-14-03932-f001:**
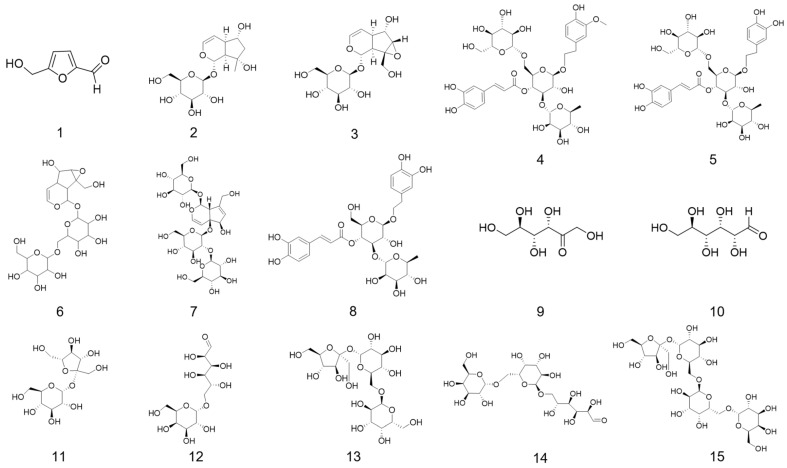
Chemical structures of small molecule ligands: 5-HMF (**1**); Aju (**2**); Cat (**3**); Cis A (**4**); Ech (**5**); Reh A (**6**); Reh D (**7**); Ver (**8**); Fru (**9**); Glu (**10**); Suc (**11**); Mel (**12**); Raf (**13**); Mnt (**14**); Sta (**15**). Note: 5-HMF, 5-hydroxymethylfurfural; Aju, ajugol; Cat, catalpol; Cis A, cistanoside A; Ech, echinacoside; Reh A, rehmannioside A; Reh D, rehmannioside D; Ver, verbascoside; Fru, fructose; Glu, glucose; Suc, sucrose; Mel, melibiose; Raf, raffinose; Mnt, manninotriose; Sta, stachyose.

**Figure 2 foods-14-03932-f002:**
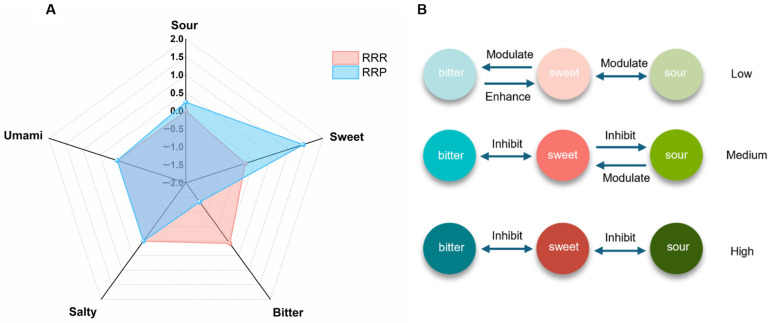
Sensory radar map of raw Rehmanniae Radix (RRR) and processed Rehmanniae Radix Praeparata (RRP) (**A**) (n = 12 panelists, independent-sample *t*-test, *p* < 0.05). Interactions between sweet, sour, and bitter across varying concentrations (**B**).

**Figure 3 foods-14-03932-f003:**
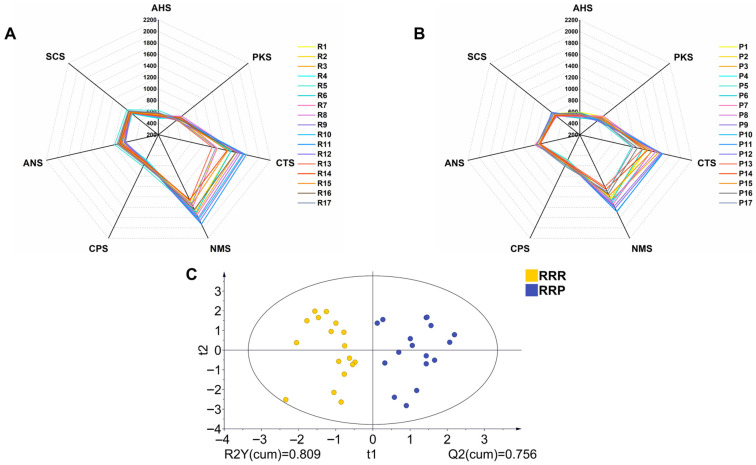
Electronic tongue radar map of RRR (**A**), and RRP (**B**) (n = 17 batches, each measured in quadruplicate after discarding first five readings). The OPLS-DA analysis (**C**) of electronic tongue. Note: R1–R17 and P1–P17 represent the 17 batches of RRR and RRP samples, respectively; AHS, sourness sensor; ANS, sweetness sensor; SCS, bitterness sensor; CTS, saltiness sensor; NMS, umami sensor; PKS and CPS, comprehensive sensor outputs.

**Figure 4 foods-14-03932-f004:**
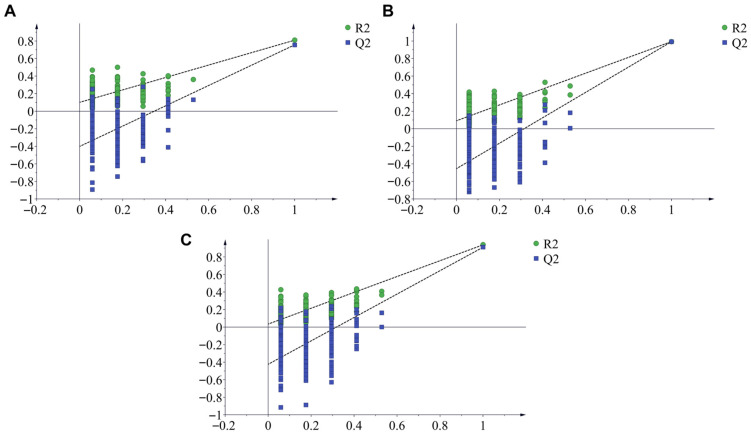
Permutation test results for OPLS-DA models based on three analytical datasets: electronic tongue response (**A**), carbohydrate contents (**B**), and marker compounds (**C**).

**Figure 5 foods-14-03932-f005:**
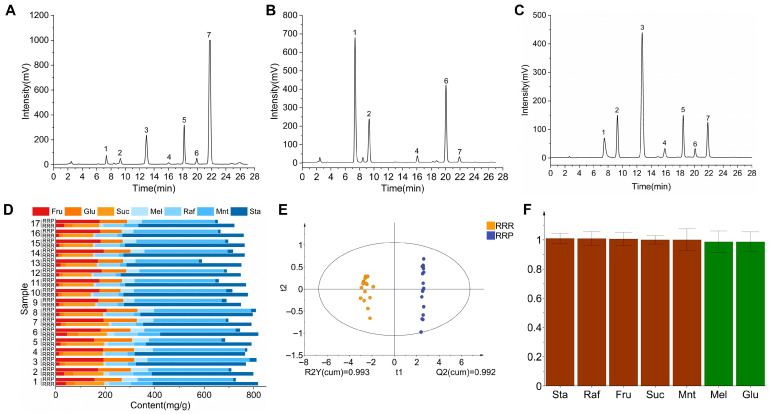
The chromatograms of HPLC-ELSD of RRR (**A**), RRP (**B**), and mixed saccharides standard (**C**); composition content analysis (**D**) (n = 17 batches, independent-sample *t*-test, *p* < 0.05); OPLS-DA analysis (**E**) and VlP values analysis (**F**). Note: 1.fructose (Fru); 2.glucose (Glu); 3.sucrose (Suc); 4.melibiose (Mel); 5.raffinose (Raf); 6.manninotriose (Mnt); 7.stachyose (Sta).

**Figure 6 foods-14-03932-f006:**
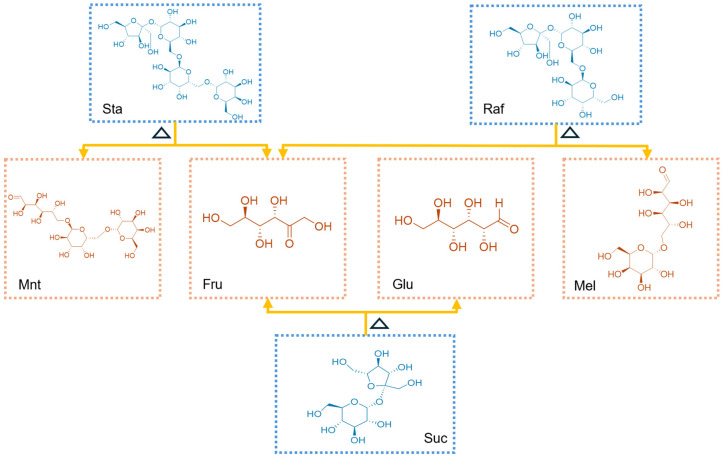
Schematic illustration of the hydrolysis mechanism of oligosaccharides under high-temperature conditions. Note: Sta, stachyose; Raf, raffinose; Mnt, manninotriose; Fru, fructose; Glu, glucose; Mel, melibiose; Suc, sucrose.

**Figure 7 foods-14-03932-f007:**
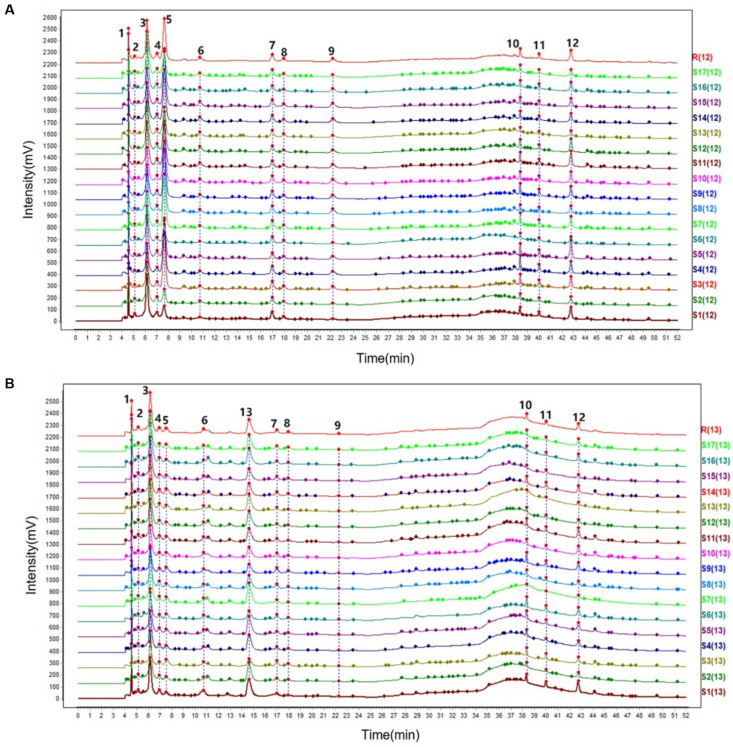
The characteristic spectrogram of HPLC-PAD of RRR (**A**) and RRP (**B**).

**Figure 8 foods-14-03932-f008:**
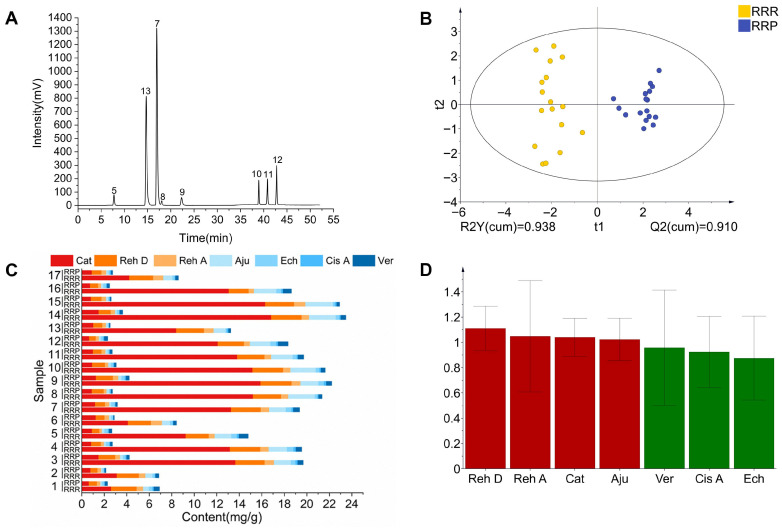
The chromatograms of HPLC-PAD of mixed marker compounds standard (**A**);composition content analysis (**B**); OPLS-DA analysis (**C**) and VlP values analysis (**D**). Note: 5.catalpol (Cat); 7.rehmannioside D (Reh D); 8.rehmannioside A (Reh A); 9.ajugol (Aju); 10.echinacoside (Ech); 11.cistanoside A (Cis A); 12.verbascoside (Ver); 13.5-hydroxymethylfurfural (5-HMF).

**Figure 9 foods-14-03932-f009:**
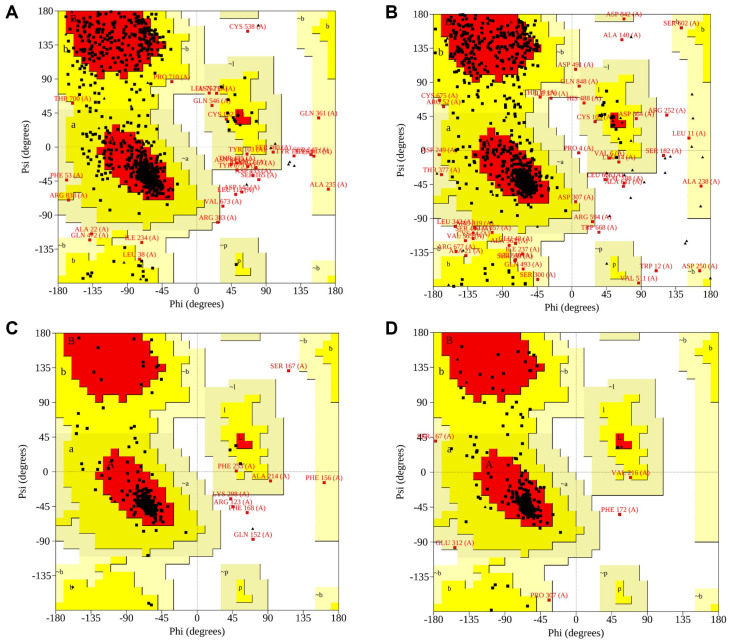
Ramachandran plots of receptor Model I: hT1R2 (**A**); hT1R3 (**B**); hT2R4 (**C**); hT2R14 (**D**). Note: Red regions: the most favored conformations; Yellow regions: the additionally allowed conformations; Black dots: the φ–ψ dihedral angle positions of individual residues.

**Figure 10 foods-14-03932-f010:**
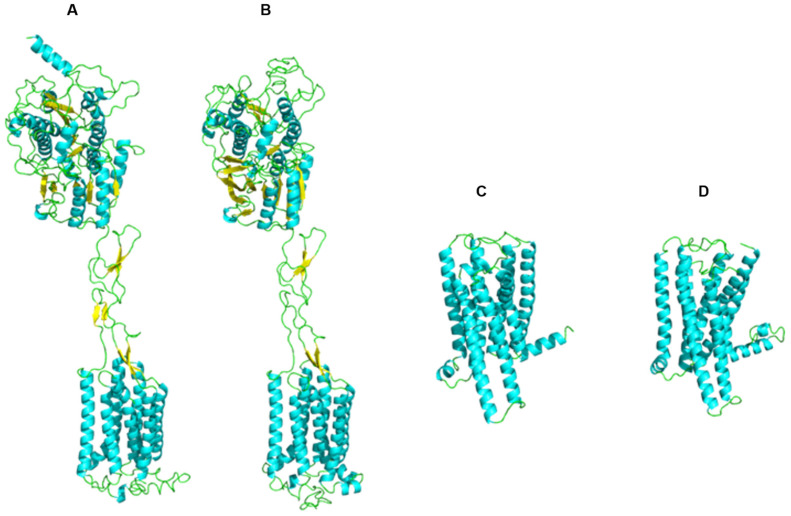
Receptor Models of hT1R2 (**A**); hT1R3 (**B**); hT2R4 (**C**) and hT2R14 (**D**).

**Figure 11 foods-14-03932-f011:**
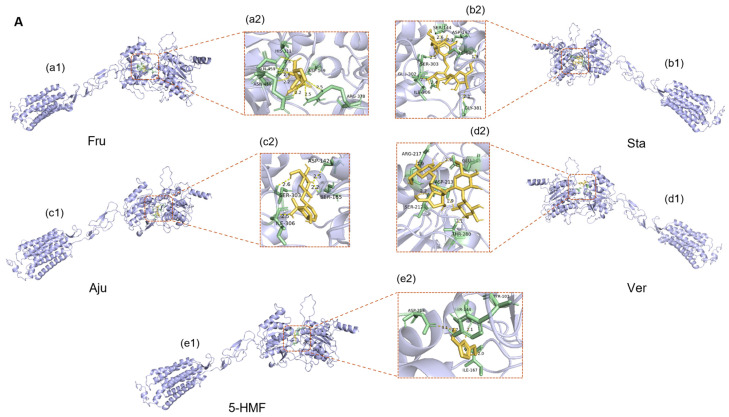
Molecular docking schematic of partial compounds (Fru, Sta, Aju, Ver, and 5-HMF) with the sweet receptor hT1R2 (**A**); molecular docking schematic of partial compounds (Fru, Sta, Aju, Ver, and 5-HMF) with the sweet receptor hT1R3 (**B**). Among these, a1–j1 represents the overall 3D structure of the protein-ligand docking pose; a2–j2 depicts the detailed 3D view of the protein-ligand interaction site.

**Table 1 foods-14-03932-t001:** Sources of Rehmannia Radix herbs of different batches and origins.

Batches	Place of Origin	Batches	Place of Origin
S1	Qu Village, Dahongqiao Township, Wuzhi County, Jiaozuo City, Henan Province	S10	Jiabu Fourth Village, Dafeng Town, Wuzhi County, Jiaozuo City, Henan Province
S2	Xitao Village, Xitao Town, Wuzhi County, Jiaozuo City, Henan Province	S11	Chenxinzhuang Village, Zhaobao Town, Wen County, Jiaozuo City, Henan Province
S3	Xitao Village, Xitao Town, Wuzhi County, Jiaozuo City, Henan Province	S12	Mafenglin Village, Wude Town, Wen County, Jiaozuo City, Henan Province
S4	Jiabu Fourth Village, Dafeng Town, Wuzhi County, Jiaozuo City, Henan Province	S13	Beibaofeng Village, Wude Town, Wen County, Jiaozuo City, Henan Province
S5	Yijing Village, Xiguo Town, Mengzhou City, Jiaozuo City, Henan Province	S14	Beibaofeng Village, Wude Town, Wen County, Jiaozuo City, Henan Province
S6	Xitao Village, Xitao Town, Wuzhi County, Jiaozuo City, Henan Province	S15	Hengshan Village, Huagong Town, Mengzhou City, Jiaozuo City, Henan Province
S7	Jiabu Third Village, Dafeng Town, Wuzhi County, Jiaozuo City, Henan Province	S16	Dangsongma Village, Gudan Town, Mengzhou City, Jiaozuo City, Henan Province
S8	Jiabu Fourth Village, Dafeng Town, Wuzhi County, Jiaozuo City, Henan Province	S17	Dangsongma Village, Gudan Town, Mengzhou City, Jiaozuo City, Henan Province
S9	Qu Village, Dahongqiao Township, Wuzhi County, Jiaozuo City, Henan Province		

**Table 2 foods-14-03932-t002:** Sensory Evaluation Scale for Rehmannia Radix.

Characters	Intensity Scale for Sensory						
		Negative		Reference		Positive	
	Strong	Moderate	Slight		Slight	Moderate	Strong
	−3	−2	−1	0	1	2	3
sweet							
sour							
bitter							
salty							
umami							
aftertaste							
others							

**Table 3 foods-14-03932-t003:** Sugar content in different processed products of Rehmanniae Radix.

Batches	Fru %	Glu %	Suc %	Mel %	Raf %	Mnt %	Sta %
R1	4.10	3.70	11.83	2.05	8.80	8.75	42.48
R2	3.38	3.73	12.20	2.40	9.85	7.33	40.92
R3	1.58	2.13	15.63	1.00	8.95	3.23	44.32
R4	1.43	1.65	15.95	1.00	9.15	2.83	44.49
R5	1.90	2.40	15.35	1.63	10.58	4.65	42.56
R6	4.55	4.58	11.60	3.08	9.50	8.98	39.58
R7	1.98	2.10	17.48	1.08	9.85	2.90	43.68
R8	1.50	1.75	15.93	1.20	11.35	2.55	45.32
R9	1.33	1.88	12.83	0.98	8.05	2.60	47.17
R10	1.08	1.45	11.70	1.19	10.05	1.95	50.20
R11	1.25	1.58	12.53	1.00	9.68	2.10	48.74
R12	1.40	1.43	11.65	0.95	8.85	2.08	48.37
R13	1.90	2.15	12.60	0.78	8.65	2.60	46.31
R14	1.08	1.58	12.48	0.87	9.58	1.85	48.86
R15	1.43	1.68	12.03	0.90	9.93	2.08	48.23
R16	1.33	1.53	12.33	0.95	8.93	1.98	48.89
R17	3.45	3.30	10.85	1.80	8.35	5.65	38.75
P1	15.70	11.03	\	6.45	\	38.63	0.91
P2	17.13	13.25	\	7.08	\	32.48	0.95
P3	19.33	12.43	\	5.43	\	41.23	2.66
P4	19.18	12.53	\	7.65	\	37.15	0.84
P5	15.58	15.28	\	6.93	\	29.28	1.36
P6	18.20	12.03	\	5.63	\	36.90	1.70
P7	19.35	13.43	\	6.83	\	29.25	0.87
P8	20.55	12.48	\	6.83	\	39.15	1.86
P9	17.20	10.10	\	4.63	\	35.25	1.81
P10	17.65	8.58	\	5.43	\	37.33	2.37
P11	17.65	9.18	\	5.15	\	32.60	1.30
P12	18.60	9.88	\	5.38	\	34.25	1.00
P13	17.05	10.23	\	5.08	\	25.60	1.07
P14	19.43	10.75	\	5.75	\	33.63	2.08
P15	18.20	8.88	\	5.53	\	35.98	1.05
P16	17.98	8.68	\	4.90	\	34.10	0.91
P17	17.70	11.18	\	6.03	\	29.60	0.95
Δavg	15.99	8.90	−13.23	4.58	−9.42	30.49	−43.83

Δavg = 1/17$\sum_{i=1}^{17}(Pi-Ri)$ × 100. Note: R1–R17 and P1–P17 represent the 17 batches of RRR and RRP, respectively; Fru, fructose; Glu, glucose; Suc, sucrose; Mel, melibiose; Raf, raffinose; Mnt, manninotriose; Sta, stachyose; Δavg. Average change rate of component content in RRP compared to RRR (%); Pi. Content of components in the i-th batch of RRP (%); Ri. Content of components in the i-th batch of RRR (%); “−”. Decrease; “\” ≈ 0, indicating trace-level content.

**Table 4 foods-14-03932-t004:** Marker compounds content in different processed products of Rehmanniae Radix.

Batches	Cat mg/g	Reh D mg/g	Reh A mg/g	Aju mg/g	Ech mg/g	Cis A mg/g	Ver mg/g
R1	2.621	2.262	0.581	0.522	0.298	0.118	0.503
R2	3.125	1.955	0.585	0.628	0.205	0.069	0.283
R3	13.664	2.601	0.828	1.491	0.427	0.178	0.488
R4	13.170	2.703	0.743	1.633	0.585	0.197	0.525
R5	4.227	2.125	0.915	0.802	0.198	0.061	0.268
R6	9.252	2.055	0.503	1.501	0.477	0.191	0.817
R7	4.115	2.039	0.977	0.717	0.246	0.052	0.266
R8	13.270	2.638	0.757	1.498	0.474	0.170	0.547
R9	15.214	2.509	0.641	2.347	0.245	0.098	0.288
R10	15.884	2.767	0.805	1.591	0.489	0.199	0.483
R11	15.174	2.722	0.650	2.265	0.321	0.125	0.383
R12	13.818	2.434	0.591	1.922	0.346	0.104	0.510
R13	12.103	2.318	0.531	1.829	0.507	0.175	0.873
R14	8.399	2.477	0.847	1.079	0.104	0.133	0.197
R15	16.848	2.707	0.664	2.365	0.340	0.107	0.450
R16	16.305	2.583	0.985	2.456	0.177	0.149	0.264
R17	13.072	1.767	0.479	1.981	0.440	0.148	0.733
P1	0.637	0.728	0.289	0.146	0.206	0.057	0.223
P2	0.776	0.636	0.303	0.154	0.135	0.034	0.103
P3	1.500	1.481	0.434	0.260	0.236	0.085	0.230
P4	0.846	0.847	0.291	0.183	0.304	0.068	0.184
P5	0.895	0.876	0.404	0.213	0.136	0.093	0.121
P6	0.906	0.665	0.237	0.219	0.301	0.100	0.254
P7	1.251	0.782	0.408	0.197	0.159	0.020	0.069
P8	1.201	0.884	0.383	0.170	0.298	0.074	0.166
P9	0.899	1.065	0.246	0.232	0.119	0.045	0.121
P10	1.268	1.518	0.475	0.277	0.298	0.102	0.275
P11	0.927	1.149	0.286	0.240	0.197	0.083	0.172
P12	0.993	0.781	0.336	0.193	0.152	0.050	0.210
P13	0.665	0.601	0.267	0.156	0.222	0.092	0.296
P14	1.025	0.817	0.316	0.186	0.055	0.053	0.078
P15	1.505	1.077	0.378	0.182	0.198	0.044	0.247
P16	0.817	0.956	0.392	0.180	0.113	0.072	0.086
P17	0.748	0.723	0.258	0.208	0.241	0.062	0.241
Δavg	−88.15	−61.96	−52.17	−84.68	−41.77	−46.94	−60.33

Δavg = 1/17$\sum_{i=1}^{17}(Pi-Ri)/Ri$ × 100. Note: R1–R17 and P1–P17 represent the 17 batches of RRR and RRP, respectively; Cat, catalpol; Reh D, rehmannioside D; Reh A, rehmannioside A; Aju, ajugol; Ech, echinacoside; Cis A, cistanoside A; Ver, verbascoside; Δavg. Average change rate of component content in RRP compared to RRR (%); Pi. Content of components in the i-th batch of RRP (mg/g); Ri. Content of components in the i-th batch of RRR (mg/g); “−”. Decrease.

**Table 5 foods-14-03932-t005:** C-score, ERRAT value, and percentage of amino acids in allowed regions of each model.

Receptor Name	Model	ERRAT Value	C-Score	Percentage of Amino Acids in Allowed Regions (%)
hT1R2	I	89.19	0.19	98.8
	II	86.10	0.02	97.6
	III	81.42	−0.83	98.1
	IV	82.13	−2.04	98.0
	V	80.34	−2.28	98.5
hT1R3	I	85.56	−0.16	98.6
	II	88.11	−0.68	97.5
	III	87.33	−1.99	97.5
	IV	84.32	−0.93	98.6
	V	82.82	−3.07	97.5
hT2R4	I	99.66	0.40	98.2
	II	92.78	−2.30	99.6
	III	98.63	−2.35	98.6
	IV	94.85	−4.20	98.9
	V	97.94	−4.58	99.6
hT2R14	I	96.76	−0.63	99.7
	II	96.76	−1.02	99.3
	III	98.63	−1.45	99.7
	IV	96.44	−3.26	99.0
	V	92.56	−3.04	97.3

**Table 6 foods-14-03932-t006:** TM-score and RMSD of receptor Model I.

Receptor Name	TM-Score	RMSD
hT1R2	0.72 ± 0.11	8.4 ± 4.5
hT1R3	0.69 ± 0.12	8.9 ± 4.6
hT2R4	0.77 ± 0.10	5.3 ± 3.4
hT2R14	0.63 ± 0.13	7.7 ± 4.3

**Table 7 foods-14-03932-t007:** Docking Energies of Compounds with Sweet (hT1R2/hT1R3) and Bitter (hT2R4/hT2R14) Taste Receptors (kJ/mol).

Compound Name	AR2	AR3	As	AR4	AR14	Ab
5-HMF	−5.1	−4.9	−10.0	−4.7	−4.9	−9.6
Aju	−8.1	−7.8	−15.9	−6.6	−6.7	−13.3
Cat	−7.8	−8.3	−16.1	−6.8	−6.6	−13.4
Cis A	−8.9	−10.8	−19.7	−7.3	−8.7	−16.0
Ech	−9.1	−9.7	−18.8	−8.0	−9.1	−17.1
Reh A	−9.8	−10.3	−20.1	−7.5	−8.1	−15.6
Reh D	−6.2	−9.3	−15.5	−6.0	−6.8	−12.8
Ver	−10.7	−9.5	−20.2	−6.9	−9.3	−16.2
Fru	−4.9	−5.5	−10.4	−5.2	−4.7	−9.9
Glu	−4.8	−5.3	−10.1	−4.9	−4.6	−9.5
Suc	−6.8	−7.1	−13.9	−5.4	−6.6	−12.0
Mel	−6.7	−7.2	−13.9	−5.9	−6.5	−12.4
Raf	−7.7	−7.8	−15.5	−5.9	−6.9	−12.8
Mnt	−7.6	−8.0	−15.6	−5.8	−6.8	−12.6
Sta	−9.5	−10.0	−19.5	−6.5	−6.8	−13.3

Note: AR2, AR3, AR4 and AR14 denote the lowest binding energies for ligand-receptor complexes of hT1R2, hT1R3, hT1R4, and hT1R14, respectively; As = AR2 + AR3; Ab = AR4 + AR14; 5-HMF, 5-hydroxymethylfurfural; Aju, ajugol; Cat, catalpol; Cis A, cistanoside A; Ech, echinacoside; Reh A, rehmannioside A; Reh D, rehmannioside D; Ver, verbascoside; Fru, fructose; Glu, glucose; Suc, sucrose; Mel, melibiose; Raf, raffinose; Mnt, manninotriose; Sta, stachyose.

**Table 8 foods-14-03932-t008:** Hydrogen bond interactions between ligands and the sweet taste receptor (hT1R2/hT1R3).

Compound Name	hT1R2 (Binding Residues: Distance, Å)	hT1R3 (Binding Residues: Distance, Å)	Compound Name	hT1R2 (Binding Residues: Distance, Å)	hT1R3 (Binding Residues: Distance, Å)
5-HMF	ASP-213:3.1; SER-144:2.1; ILE-167:2.0; TYR-103:2.7	SER-182:2.1; PHE-183:2.4	Fru	HIS-311:2.4; GLN-459:2.1,2.5; ASN-460:2.2; ASP-169:2.2; ARG-378:2.5,2.5	GLN-193:2.8; GLY-168:2.4,2.5; GLU-301:2.4
Aju	ASP-142:2.5; SER-303:2.6; SER-165:2.2; ILE-306:2.5	HIS-145:2.5; GIU-301:2.3,2.5; GLN-389:2.3	Glu	HIS-311:2.7; ASN-460:2.4,2.5, 2.6; ARG-378:2.5,2.6	ALA-302:2.2; SER-306:2.4,2.8; THR-305:2.0,2.1; HIS-388:2.1
Cat	GIU-302:2.3; ILE-306:2.8	ASP-307:2.8; SER-306:2.4; THR-305:2.3; HIS-388:2.7	Suc	SER-303:2.0,2.3; ILE-306:2.0	HIS-278:2.3
Cis A	TYR-62:2.6; ASN-312:2.2; VAL-64:2.2; CLN-355:2.3, 2.4; SER-356:2.5, 1.8; ASN-374:2.2; SER-372:2.4	GLU-45:2.4; SER-170:2.7; TRP-72:2.0; HIS-388:2.6; ASN-386:2.3,2.4	Mel	SER-144:2.5; ASP-142:2.4,3.5; SER-165:2.1; ILE-306:2.0	HIS-145:2.1,2.3; ASN-68:2.0; ASP-307:2.2; THR-305:2.6; HIS-388:2.4
Ech	ASP-278:2.3, 2.4; ILE-167:2.6,2.9; SER-165:2.9; GLU-302:2.4	ASP-307:2.1,2.6; THR-305:2.0; GLU-45:2.1; SER-67:2.3; TRP-72:1.8	Raf	ASP-142:2.6, 2.8; SER-165:2.4; ILE-306:2.2; VAL-384:2.8	SER-146:1.9; GLY-44:2.3; VAL-277:2.6; ALA-302:2.2; ASN-68:1.8
Reh A	SER-40:2.4; ILE-306:2.4; GLU-302:2.2; ARG-378:2.4, 2.5,2.6	GLY-44:2.2; VAL-277:2.8; HIS-145:2.7; ASN-68:2.5,2.7; THR-305:1.8,2.4; ASN-386:1.9,2.2	Mnt	SER-144:2.4; ASP-142:2.7	SER-147:2.5; SER-104:2.5; SER-146:2.2; PRO-42:2.6; SER-66:2.5; THR-305:2.2,2.4; HIS-388:2.7
Reh D	ARG-217:1.9; GLU-145:2.3; ASP-213:1.9; SER-212:2.2; THR-280:2.1	HIS-278:2.1; SER-147:2.2; HIS-145:1.9,2.4; SER-306:2.7; HIS-388:2.6	Sta	SER-144:2.6; ASP-142:2.3; ILE-67:1.9; SER-303:2.5; GLU-302:2.4; ILE-306:2.3; GLY-381:2.1	SER-104:2.1; SER-146:1.9,2.6; TYR-218:2.8; HIS-145:1.9,2.2; GLN-389:2.2,2.6; ASN-68:2.0; HIS-388:2.2,2.4, 2.6; ASP-307:2.5
Ver	SER-165:2.1; ASN-70:2.3,2.5	ASN-68:2.0,2.5; ASN-386:2.8			

Note: 5-HMF, 5-hydroxymethylfurfural; Aju, ajugol; Cat, catalpol; Cis A, cistanoside A; Ech, echinacoside; Reh A, rehmannioside A; Reh D, rehmannioside D; Ver, verbascoside.

## Data Availability

The original contributions presented in the study are included in the article/Appendix A, further inquiries can be directed to the corresponding authors.

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
