# Peer review of "Chemical Mechanisms Underlying Sweetness Enhancement During Processing of Rehmanniae Radix: Carbohydrate Hydrolysis, Degradation of Bitter Compounds, and Interaction with Taste Receptors"

_foods, 2025, doi:10.3390/foods14223932_

Round 1
Reviewer 1 Report
Comments and Suggestions for Authors
This manuscript mainly report the mechanism study of sweetness enhancement during processing of Rehmanniae Radix. Using RRR and RRP as the material, the enhancement of the sweetness was confirmed by both sensory test and electronic tongue analysis initially. Later, chemical analysis was focused on the sugars, and a systematic comparison was made. Finally, a molecular docking study was applied to predict the activator and receptor. Overall, this is pretty interesting and significant study. The story is logic, and the experiment design is good as well. As far as I am concerned, this manuscript can be assigned as minor revision.
Here are my questions and concerns.
1) In the introduction part, the chemical composition of Rehmanniae Radix from literature is missing. It is good to mention the bioactive component as the secondary metabolite, or it contains decent amount of light polysaccharide, which can be easily composed to mono or di saccharide. I don’t think all the Chinese medicine has this processing-based sweetness enhancement property; there must be some uniqueness for Rehmanniae Radix.
2) This manuscript attributes to the sweet enhancement mainly to the simple sugar. However, besides that, any other phytochemical may also be responsible for that? Under the processing, is there any other significant component change, besides the sugar composition between RRR and RRP?
Author Response
We sincerely thank the editors and reviewers for their constructive comments and valuable suggestions.
To provide a clear and comprehensive response, we have prepared a separate document entitled “Response to Reviewers”, in which all comments are addressed point by point, with corresponding revisions highlighted in the revised manuscript.
Please kindly refer to the attached file for the full response.

Reviewer 2 Report
Comments and Suggestions for Authors
The article entitled: "Chemical Mechanisms Underlying Sweetness Enhancement During Processing of Rehmanniae Radix: Carbohydrate Hydrolysis, Degradation of Bitter Compounds, and Interaction with Taste Receptors" studies the different characteristics of Raw Rehmannia Radix and Rehmannia Radix Praeparat from a chemical-physical and sensorial point of view.
The article is well presented and certainly similar to the journal scope.
However, it does not bring any knowledge, innovation or novelty. The differences of the two different "products" are reported, essentially due to the process to which the root is subjected. These changes are quite well known and reported in the literature. According to the authors, "the results also provide a scientific basis for improving the sensory quality of RRP" but it would be interesting to conduct transformation tests and investigate the improving effects compared to the product known to date.
Author Response

(The authors gave the same response as above.)

Reviewer 3 Report
Comments and Suggestions for Authors
The authors wrote a comprehensive introduction covering the topics of the presented research. They explained why this research topic was chosen and described the molecular basis for the bitter and sweet taste of the product. The introduction concludes with a clarification of the three ambitious goals of the study.
Line 126- please explain here the abbreviations “RRR and RRP slices”
Line 126-127- The authors must describe in detail the processes used to treat the research material described in the Chinese Pharmacopoeia. This is not a publicly available document, and the processes to which the Rehmannia Radix samples were subjected are unique and crucial to the entire article.
Line 218-219- please describe how you obtained the coarse powder of RRR and RRP samples, with providing model name of the device use for this purpose
2.4.2. Characteristic Spectrum and Marker Compound Analysis - Authors should add a comment specifying what kind of substances they are analyzing and what they are looking for using this methodology.
The description of statistical analyses and the selection of methods is appropriate.
Figure 3- All parts A, B, and C should be better described in the text, and the footer of the figure should include explanations of what the abbreviations refer to, e.g., in radar charts.
Table 3 – why sugar content is not expressed in quantity values, like mg/kg? Authors should indicate in the footer what the P and R symbols with numbers refer to.
Figure 5D, 8C – should be larger, it's hard to see anything on these figures
Line 400-401, Table 4- The authors did not write the full names of the compounds anywhere: Peak 5 (Cat), Peak 7 (Reh D), Peak 8 (Reh A), Peak 9 (Aju), 401 Peak 10 (Ech), Peak 11 (Cis A), Peak 12 (Ver), and Peak 13 (5-HMF).
The authors presented conclusions consistent with the data presented, and these conclusions are original. However, there is no mention of what processes caused the effect mentioned by the authors.
Author Response

(The authors gave the same response as above.)

Round 2
Reviewer 2 Report
Comments and Suggestions for Authors
I sincerely appreciate the improvement in manosctict autorei in general. Furthermore, the highlighted connection between the mechanistic aspects that were previously little emphasized is appreciable, giving greater prominence to the work conducted than what is present in the literature.
Reviewer 3 Report
Comments and Suggestions for Authors
The authors responded to all comments and remarks. They supplemented key information on sample preparation and other information on the methodology of the experiments. In addition, they improved the graphic quality and descriptions of graphs and tables. The authors referred to the graphs to a greater extent in the discussion of the results. The conclusions chapter has also been improved, which clearly allows the conclusions of the authors' research to be extracted and read.